# A Case of Class I 17p13.3 Microduplication Syndrome with Unilateral Hearing Loss

**DOI:** 10.3390/genes14071333

**Published:** 2023-06-24

**Authors:** Spiros Vittas, Maria Bisba, Georgia Christopoulou, Loukia Apostolakopoulou, Roser Pons, Pantelis Constantoulakis

**Affiliations:** 1MicroGenome, 25th Martiou 55 Str., 564 29 Thessaloniki, Greece; mpismpamaria@gmail.com; 2Genotypos Science Labs Medical SA, 3-5 Ilision Str., 115 28 Athens, Greece; gchristopoulou@genotypos.gr (G.C.); pconstantoulakis@genotypos.gr (P.C.); 3First Department of Pediatrics, National and Kapodistrian University of Athens, Aghia Sofia Hospital, 115 27 Athens, Greece; loukiaap@med.uoa.gr (L.A.); roserpons@med.uoa.gr (R.P.)

**Keywords:** 17p13.3, microduplication, *YHWAE*, array-CGH, WES, neurodevelopmental disorders

## Abstract

17p13 is a chromosomal region characterized by genomic instability due to high gene density leading to multiple deletion and duplication events. 17p13.3 microduplication syndrome is a rare condition, reported only in 40 cases worldwide, which is found in the Miller–Dieker chromosomal region, presenting a wide range of phenotypic manifestations. Usually, the duplicated area is de novo and varies in size from 1.8 to 4.0 Mbp. Critical genes for this region are *PAFAH1B1* (#601545), *YWHAE* (#605066), and *CRK* (#164762). 17p13.3 microduplication syndrome can be categorized into two classes (Class I and Class II) based on the genes that are present in the duplicated area, which lead to different phenotypes. In this report, we present a new case of Class I 17p13.3 microduplication syndrome that presents with unilateral sensorineural hearing loss. Oligonucleotide and SNP array comparative genomic hybridization (a-CGH) analysis revealed a duplication of approximately 121 Kbp on chromosome 17p13.3, which includes *YWHAE* and *CRK* genes. Whole-exome sequencing (WES) analysis confirmed the duplication. Our patient has common clinical symptoms of Class I 17p13.3 microduplication syndrome, and in addition, she has unilateral sensorineural hearing loss. Interestingly, WES analysis did not detect any mutations in genes that are associated with hearing loss. The above findings lead us to propose that hearing loss is a manifestation of 17p13.3 duplication syndrome.

## 1. Introduction

Chromosome 17p13.3 is characterized by genomic instability due to the presence of high-density low-copy repeats that are associated with susceptibility to submicroscopic rearrangements leading to microdeletion and microduplication syndromes [1]. 

Chromosome microdeletions within the 17p13.3 chromosomal region have been previously recorded and result in either isolated lissencephaly spectrum disorder (isolated lissencephaly sequence—ILS) (OMIM: #607432) or Miller–Dieker lissencephaly syndrome, which is also known as chromosome 17p13.3 deletion syndrome (OMIM: #247200), because of either a heterozygous mutation in the *PAFAH1B1* gene or an insufficiency of the *PAFAH1B1* gene, respectively [2]. Clinical characteristics of these syndromes are typical of lissencephaly (pachygyria and the incomplete or absent gyration of the cerebrum), microcephaly, hypotonia, dysmorphic facial features (narrow forehead, downward-slanting palpebral fissures, or small nose and chin), cardiac malformations, growth retardation, and mental deficiency with seizures and EEG abnormalities. On the other hand, the overexpression of *PAFAH1B1* leads to a recently discovered condition known as 17p13.3 microduplication syndrome (OMIM #613215) with a wide range of phenotypic manifestations [1,3]. Despite the fact that both types of copy number variations share common phenotypic features, they present distinct phenotypes. Bruno L.D. et al., 2010 [4], support that intrauterine growth retardation, developmental delay, special facial features, and structural brain abnormalities are some of the manifestations that can be found in both conditions, but behavioral problems and autistic spectrum disorders are observed only in duplications. Based on the literature, congenital malformations, hypotonia, and subtle hand/foot malformations are also manifestations associated with the duplication [4,5]. Usually, the duplicated area is de novo and varies in size from 1.8 to 4.0 Mb [6,7], although cases with smaller duplicated areas have also been reported [8]. 

Clinical manifestations of 17p13.3 microduplication syndrome depend on the gene content of the duplicated region. Critical genes in this region include *PAFAH1B1, YWHAE,* and *CRK.* The overexpression of the *PAFAH1B1* gene, encoding LIS1, has been shown to affect brain development by causing migration defects. *YWHAE* encodes 14-3-3*ε*, for which dosage alterations are suggested to affect neuronal network development and maturation, and *CRK* is known to interact with signal pathways involved in brain and limb development [9].

17p13.3 microduplication syndrome can be categorized into two classes (Class I and Class II). Class I duplications involve *YWHAE* but not *PAFAH1B1*, and the phenotype includes autistic features, speech and motor delay, and subtle facial and hand/foot dysmorphisms. Class II duplications always include the *PAFAH1B1* gene and may also include *YWHAE* and *CRK* with a phenotype characterized by developmental delay, growth restriction, microcephaly, and other brain malformations such as the hypoplasia of the corpus callosum and mild cerebellar volume loss [1,8]. According to the literature, *PAFAH1B1* and *YWAHE* have been used so far in order to classify patients carrying a duplication in 17p13.3 in either Class I or II and, therefore, to predict the resulting distinct phenotype. On the contrary, recent studies support that, in cases of 17p13.3 microdeletion syndrome, *PAFAHB1* is considered necessary for lissencephaly’s phenotype of Miller–Dieker patients, and the presence of *YWHAE* increases the severity of the manifestations, which highlights the differences between the two conditions (duplications vs. deletions). 

Additionally, cases that carry deletions in the 17p13.3 chromosomal region seem to be easier to detect prenatally during ultrasound examination, with corpus abnormalities being part of the main findings. On the other hand, cases carrying a microduplication of the region are often a random finding during screening, making the prenatal diagnosis of this condition a challenging procedure [10]. Furthermore, the genetic counseling is even more complicated as microduplication’s phenotype expressivity and penetrance are not predicted.

To date, only 40 cases of 17p13.3 microduplication syndrome have been reported worldwide [1], with the first case ever reported in 2009 [6,8] to the best of our knowledge. In this report, we present a new case of Class I 17p13.3 microduplication syndrome, which presented with unilateral sensorineural hearing loss, a clinical manifestation not previously described in the literature that expands the phenotypic spectrum of this disorder.

## 2. Clinical Report

This 4-year-old girl is the first child of a healthy non-consanguineous couple. During pregnancy, the proband’s mother received thyroid hormone supplementation. She was not exposed to drugs, toxins, or teratogens. The proband’s fetal growth was delayed, and a single umbilical artery was noted. She was born via vaginal delivery at 35 weeks gestation. Birth weight was 2379 g, height was 45 cm, and head circumference was 32 cm. Abnormal automated otoacoustic emissions were noted on the left side, and subsequent brainstem auditory-evoked responses confirmed left-sided sensorineural hearing loss (Table 1). 

During infancy, the proband was noted to be hypotonic, and her psychomotor development was delayed. At the age of 6 months, she started having stereotypic movement of the head. She walked at 18 months of age and said her first words at 2 years. She has worn a hearing aid since she was 2 ½ years old and uses corrective eyeglasses for hypermetropia. A brain MRI at the age of 2 ½ years showed no evidence of brain dysplasia. A small non-enhancing area of an increased signal in the left occipital white matter was noted that was interpreted as a nonspecific gliotic change. An assessment of other system involvement (heart, kidneys, and endocrine system) did not show associated abnormalities.

Now, at 4 years old, poor gross motor skills have been noticed, and she falls easily. At this age, she has the ability to go up and down stairs with support but is not able to run. The proband was toilet trained since the age of 3 ½ years. Concerning the child’s speech and language skills, she knows multiple words and can make sentences up to four words long, follows two-step commands, and follows a simple conversation when prompted by a grownup. She is also able to perform age-appropriate activities of daily living. Additionally, she has some stereotypic behaviors including head movements, shuddering, and a tendency to grind her teeth when excited. On exam (at 4 years old), the proband’s weight was 13 kg, her head circumference was 47.5 cm, and her height was 96 cm (Table 1). She has mild dysmorphic features including a flattened midface, a flat nasal bridge, and pes planus. She is happy and cooperative and has head stereotypies (side-to-side and ear-to-shoulder movements). Her cranial nerve exam is within normal limits. Nevertheless, she has generalized hypotonia and joint hypermobility, and her muscle strength is preserved. Her deep tendon reflexes are present and symmetric, and her plantar reflexes are downgoing. She shows evidence of dyspraxia in her hand movements and action tremors during finger-to-nose testing. Her gait is clumsy (Table 1).

## 3. Cytogenetic and Molecular Studies 

Whole genomic DNA was extracted from the probands’ peripheral blood with magnetic bead separation using an automated extractor device and used for oligonucleotide and SNP array comparative genomic hybridization (a-CGH) analysis and whole-exome sequencing (WES) analysis. a-CGH analysis was performed using the Affymetrix Cytogenetics Whole-Genome CytoScan 750K array platform. The results were analyzed using the Chromosome Analysis Suite Software (ChAS ver3.1 Affymetrix, Thermo Fisher Scientific, Waltham, MA, USA) according to human genome assembly GRCh37:Feb.2009 hg19. The analysis revealed a 121 kbp duplication on chromosome 17 (ISCN 2016: arr[GRCh37] 17p13.3(1203449_1324017)×3), which included three OMIM-listed genes, including *RPH3AL* (#604881), *TRARG1* (#612211), and *YWHAE* (#605066), and parts of *TUSC5* (#612211) and *CRK* (#164762) (Figure 1).

Conventional karyotype analysis was also performed on the probands’ peripheral blood showing no structural abnormalities that could lead to the gain or loss of genetic material (Figure 2).

The above findings were subsequently confirmed by the whole-exome sequencing analysis (WES) of 21.285 human genes using the Twist Human Core Exome EF Multiplex Complete kit from TWIST Bioscience carried on NextSeq500 by Illumina. Next-generation sequencing data analysis was performed by pipelines that were validated by Sophia DDM (Sophia Genetics). Copy number variation analysis revealed three copies (duplication) on chromosome 17 (17p13.3) with genomic coordinates GRCh37:chr17:(1201584_1248716) - (1303429_1326781). The duplication was estimated to be 55–130 kbp, including at least *YWHAE* (OMIM #605066) (Figure 3). 

Chromosomal imbalances in a proband can be created de novo or can be the result of either a chromosomal imbalance (duplication or deletion) or a structural chromosomal abnormality (translocation or inversion) in one of the biological parents of the proband. In order to investigate the reason for the chromosomal duplication found in our case, whole genomic DNA from peripheral blood samples of the probands’ biological parents was also used for a-CGH and WES analyses. Conventional karyotype analysis was also performed on the probands’ biological parents in order to determine whether the cause of the duplication was a balanced translocation in one of the proband’s parents. All the above analyses revealed that both biological parents do not carry any balanced or unbalanced chromosomal abnormality, and, thus, the duplication in the proband was created de novo (Figure 4 and Figure 5).

## 4. Discussion

17p13.3 microduplication syndrome (OMIM #613215) is detected inside the Miller–Dieker region on chromosome 17 and is a rare disorder with a worldwide prevalence of less than 1/1,000,000 (ORPHA: 217385). A wide range of clinical manifestations has been reported in association with this syndrome, which appears to be related to the size of the duplication, the critical genes involved, and the breakpoints of the duplication. Yet, the clinical manifestations are not pathognomonic, and it is unlikely that physicians will recognize this syndrome based on clinical grounds alone [11]. 

17p13.3 microduplication syndrome can be divided into class I and class II based on the absence or presence of *PAFAH1B1*, respectively, in the duplicated area. Although the two classes share common phenotypic features including developmental and behavioral abnormalities, phenotype differences can be detected. Class I patients usually show behavioral characteristics of ASD (a feature that is not detected in class II patients), learning disabilities, craniofacial dysmorphic features, hand/foot malformations, and growth developmental anomalies. Class II patients tend to show psychomotor developmental delay with associated hypotonia, microcephaly, and mild behavioral problems, which makes them recognized as more open to developing relationships with other people. It has been pointed out that both classes have variable expressivity and incomplete penetrance, showing remarkable phenotypic differences between patients with common genotyping characteristics [1,8,11,12]. It seems that *YWHAE*, which can be present in both classes, is responsible for attention deficit hyperactivity disorder (ADHD) and cleft lip and palate, making the investigation of phenotype–genotype correlations between the two classes more difficult [1]. 

The patient presented in this report carried a duplication that included *YWHAE* but not *PAFAH1B1*, leading us to the hypothesis that she is a case of Class I microduplication syndrome because of the critical genes involved in the duplicated area. Based on our findings, the duplication was created de novo. According to the literature, the majority of the reported cases of Class I duplications were not inherited. In some patients, the duplication was inherited by their mother, but there is no evidence so far for paternal inheritance. In these cases of the parental inheritance of the duplication, it is referred that the parental phenotype was milder than the proband’s [12]. In our case, the patient’s examination revealed that her main clinical findings include hypotonia, global developmental delay, mild facial dysmorphic features, and stereotypic movements of her head. She also has joint hypermobility, which is strongly believed to be associated with ASD, making it a subgroup of manifestations of the ASD phenotype [13]. Her presentation is consistent with Class I 17p13.3 microduplication syndrome with a milder dysmorphic and neurodevelopmental phenotype. In addition, our patient has unilateral sensorineural hearing loss, a feature that has not been previously described in Class I 17p.13.3 microduplication [5]. Only a single patient with hearing loss carrying Class II duplication that involved most of the 17p13.3 region (2.16 Mbp), described by Curry et al, has been reported so far. More specifically, a single 14-year-old patient with hearing loss, mild intellectual disability, and hypotonia was found to carry this duplication that has been associated with the probands’ phenotype. Interestingly, in this report, the type of hearing deficit was not specified, but the patient suffered from recurrent bilateral otitis media that may have played a role in the patient’s hearing impairment [5]. 

To date, more than 150 genes have been associated with hearing loss. Syndromic hearing loss is frequently associated with chromosome micro-imbalances, while in non-syndromic hearing loss, chromosomal alterations tend to be rare and small and involve specific genes such as *GJB2, MYH9, OTOA, PCDH15, SLC26A4, STRC, TMC1, TMPRSS3,* and *USH2A* [14]. Hearing loss of genetic origin is often bilateral, although unilateral involvement can also occur [15,16,17]. In a large series of patients with non-syndromic hearing loss, only 1% of those with unilateral involvement showed a genetic diagnosis that corresponded to one patient with Branchio-oto-renal syndrome, a neurocristopathy characterized by the abnormal development of the second branchial arch due to mutations in the *EYA1* gene [15]. In a series of 423 children with sensorineural hearing loss, 123 had unilateral involvement. In 13% of these patients with unilateral involvement, a genetic etiology was suspected. In 5%, mutations in the *GJB2/6* genes were found, and in 25%, congenital structural anomalies of the temporal bone were detected [16]. In a recent series, 16.8% of patients with hearing loss had unilateral involvement, and 3% of them had a genetic diagnosis [17].

In our reported patient, sensorineural hearing loss was not associated with structural abnormalities of the temporal bone. Whole-exome sequencing did not show any pathogenic or likely pathogenic variants in known genes associated with hearing loss, and array CGH did not show the presence of other CNV. Consequently, despite the rarity, we believe that it is highly possible that the unilateral hearing loss in our patient is a manifestation of 17p13.3 microduplication, and we propose that this condition should be considered in the assessment of patients with unilateral sensorineural hearing loss. 

In conclusion, we presented a case of class I 17p13.3 microduplication syndrome with a variety of symptoms. Our patient also carried unilateral sensorineural hearing loss, which is a new manifestation of the syndrome, but the exact phenotype–genotype correlations were not predicted, a fact that makes it necessary to establish specific guidelines to create more comprehensive genetic counseling for each case.

## Figures and Tables

**Figure 1 genes-14-01333-f001:**
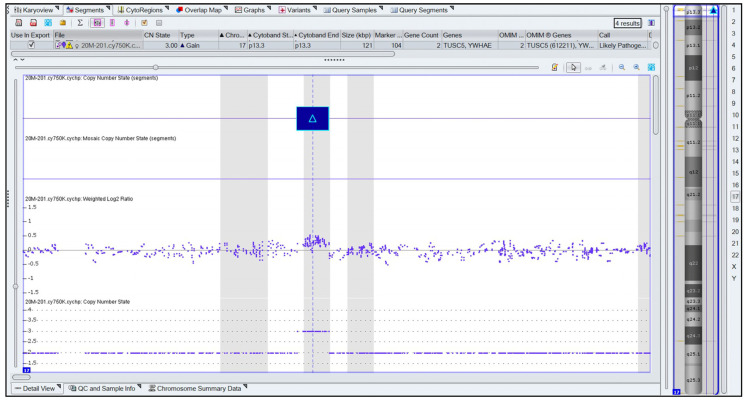
Array-CGH of the proband. Array-CGH analysis conducted using Affymetrix Cytogenetics Whole-Genome CytoScan 750K array platform according to human genome assembly GRCh37:Feb.2009 hg19. Blue box indicates the duplicated area. Array probes’ weighted log2 ratio and copy number state are also shown. Chromosomal region of the duplication is shown on the table above and on the karyogram on the right.

**Figure 2 genes-14-01333-f002:**
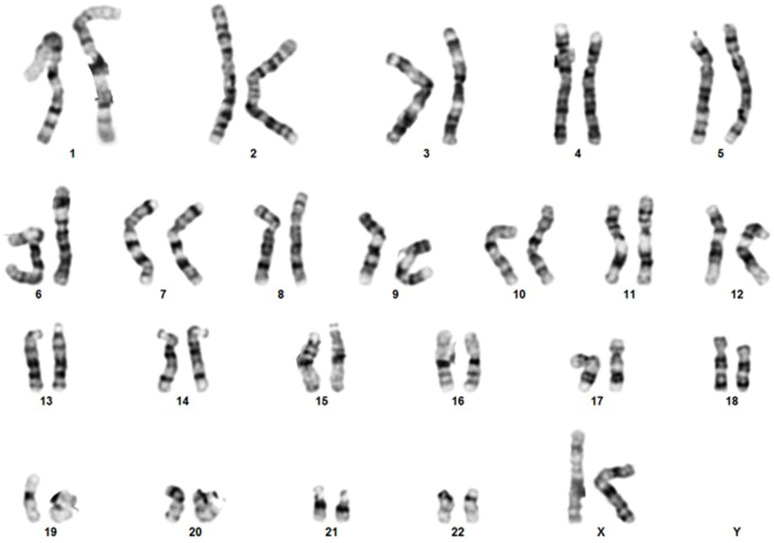
Cytogenetic analysis of the proband. Chromosomal analysis of peripheral blood leukocytes at banding quality of 400 bands per haploid set revealed no structural abnormalities.

**Figure 3 genes-14-01333-f003:**
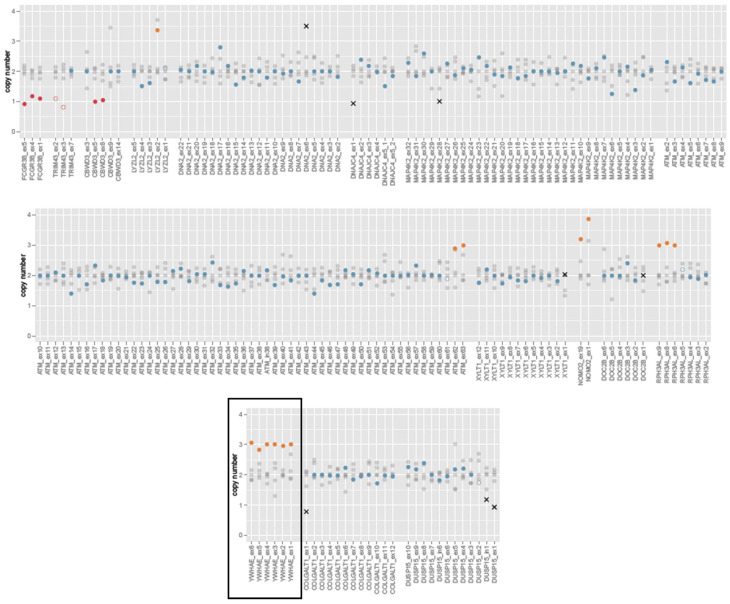
Whole-exome sequencing of the proband’s DNA. 21.285 human genes were sequenced using Twist Human Core Exome EF Multiplex Complete kit of TWIST Bioscience carried on NextSeq500 by Illumina. Analysis performed by pipelines validated by Sophia DDM. Black box indicates the area of *YWHAE* gene that is indicative of the duplicated area (orange circles).

**Figure 4 genes-14-01333-f004:**
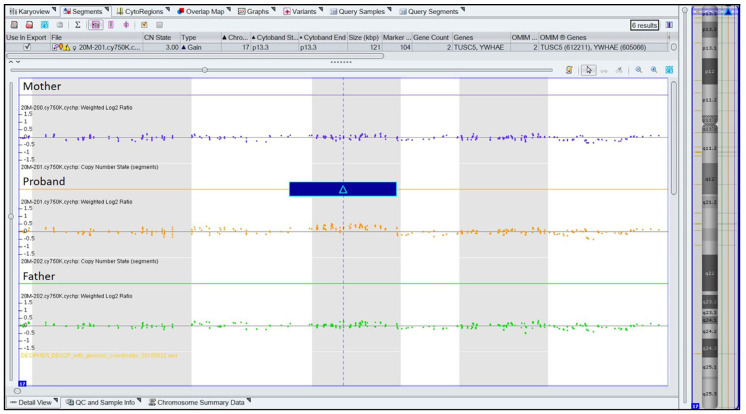
Comparative representation of the duplicated area between proband and her parents. Array-CGH analysis conducted using Affymetrix Cytogenetics Whole-Genome CytoScan 750K array platform according to human genome assembly GRCh37:Feb.2009 hg19. Blue box indicates the duplicated area in the proband only.

**Figure 5 genes-14-01333-f005:**
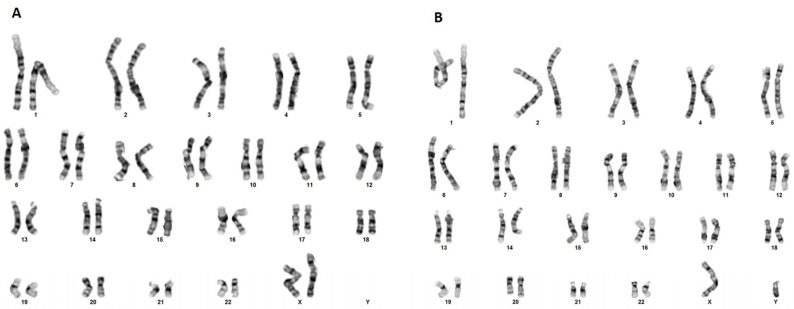
Cytogenetic analysis on peripheral blood leucocytes of proband’s mother (**A**) and father (**B**), indicating that there are no structural anomalies that could cause the duplication and, thus, classifying proband’s finding as de novo.

**Table 1 genes-14-01333-t001:** Main clinical findings at birth and early childhood.

**Somatometric Parameters**		**Percentile**
At birth		
Weight	2379 g	<3rd
Height	45 cm	<3rd
Head circumference	32 cm	5th < *p* < 15th
At examination (4 years old)		
Weight	13 Kg	5th < *p* < 15th
Height	96 cm	5th < *p* < 15th
Head circumference	47.5 cm	5th < *p* < 15th
**Clinical findings**		
At birth—infancy		
Abnormal automated otoacoustic emissions		
Hypotonia		
Delayed psychomotor development		
At examination (4 years old)		
Poor gross motor skills		
Mild dysmorphic features (flattened midface, flat nasal bridge, and pes planus)		
Generalized hypotonia and joint hypermobility		
Hypermetropia		
Left-sided sensorineural hearing loss		

## Data Availability

Data sharing not applicable.

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
