# Peer review of "A Case of Class I 17p13.3 Microduplication Syndrome with Unilateral Hearing Loss"

_genes, 2023, doi:10.3390/genes14071333_

Round 1

Reviewer 1 Report

I have the following comments:

The use of multiple genetic analysis techniques, including oligonucleotide and SNP array comparative genomic hybridization (a-CGH) and Whole Exome Sequencing (WES), adds strength to the study's findings. The confirmation of the duplication on chromosome 17p13.3, involving the YWHAE and CRK genes, through these techniques enhances the credibility of the reported case.

The authors appropriately categorize the 17p13.3 microduplication syndrome into two classes based on the genes present in the duplicated area. This classification helps explain the different phenotypic manifestations observed in affected individuals.

The observation of unilateral sensorineural hearing loss in the reported case is intriguing. The authors discuss the absence of mutations in known hearing loss genes, suggesting that the hearing impairment may be directly linked to the 17p13.3 duplication. This finding adds to the understanding of the phenotypic spectrum of the syndrome and warrants further investigation into the mechanisms underlying this association.

Thus, I aggree the acceptance of the paper and suggest more discription on the known function, expression, and related reports of the two genes within the duplicate.

Reviewer 2 Report

The paper is interesting and brings new information.

Line 47: add year of publication.

Line 70: delete space before expression “In this report.” And line 100, 101, 193, 207, 211, 214.

Table 1: add the age next to the expression “At examination”.

In the section “Clinical report”: modify the spelling to remove some of the "she", especially in the third paragraph.

Figure 1: It is not good. Maybe cut leaving more focus on the duplicated region. Perhaps removing the karyotype image and leaving only the array.

I suggest deleting the second figure 3. It is unnecessary.

In the section “Clinical report”: modify the spelling to remove some of the "she", especially in the third paragraph.
